# Vaccine Effectiveness against SARS-CoV-2 Infection during the Circulation of Alpha, Delta, or Omicron Variants: A Retrospective Cohort Study in a Tertiary Hospital in Serbia

**DOI:** 10.3390/vaccines12020211

**Published:** 2024-02-18

**Authors:** Danijela Đurić-Petković, Vesna Šuljagić, Vesna Begović-Kuprešanin, Nemanja Rančić, Vladimir Nikolić

**Affiliations:** 1Institute of Microbiology, Military Medical Academy, 11000 Belgrade, Serbia; danijeladjuricpetkovic@gmail.com; 2Department of Healthcare-Related Infection Control, Military Medical Academy, 11000 Belgrade, Serbia; 3Medical Faculty, Military Medical Academy, University of Defence, 11000 Belgrade, Serbia; begovickupresanin@hotmail.com (V.B.-K.); nece84@hotmail.com (N.R.); 4Clinic for Infectious and Tropical Diseases, Military Medical Academy, 11000 Belgrade, Serbia; 5Centre for Clinical Pharmacology, Military Medical Academy, 11000 Belgrade, Serbia; 6Institute of Epidemiology, Faculty of Medicine, University of Belgrade, 11000 Belgrade, Serbia; vladimir.nikolic@med.bg.ac.rs

**Keywords:** COVID-19, vaccination effectiveness, hospitalized patients, vaccine combinations, SARS-CoV-2 variants

## Abstract

The COVID-19 pandemic prompted rapid vaccine development and deployment worldwide. Despite widespread vaccination efforts, understanding the effectiveness of vaccines in hospitalized patients remains a critical concern. This retrospective cohort study, conducted at a tertiary healthcare centre in Serbia, tracked patients hospitalized during different waves of COVID-19 variants—Alpha, Delta, and Omicron. Data collection included demographics, comorbidities, symptoms, and vaccination status. Among 3593 patients, those with prior exposure to COVID-19 cases or hospital treatment showed higher positivity rates. Symptom prevalence varied across waves, with coughs persisting. Patients without chronic diseases were more frequent among those testing negative. Vaccine effectiveness varied, with Sinopharm demonstrating a 45.6% effectiveness initially and Pfizer-BioNTech showing an effectiveness of up to 74.8% within 0–84 days after the second dose. Mixed-dose strategies, notably Sinopharm as a primary dose followed by a Pfizer-BioNTech booster, suggested increased protection. Despite substantial vaccination availability, a significant portion of hospitalized patients remained unvaccinated. This study underscores the dynamic nature of vaccine effectiveness and advocates for booster strategies to address evolving challenges in combating COVID-19, particularly in hospitalized patients.

## 1. Introduction

At the end of 2020, after numerous experimental studies [1,2,3,4], different types of vaccines against the coronavirus disease of 2019 (COVID-19) became available in the wider population, as one of the most significant and effective measures to prevent COVID-19. Then, real-life observational studies showed the protective effect of vaccines in different parts of the world and in different populations [5,6,7,8,9]. The widespread uptake of COVID-19 vaccines has become the most powerful weapon in controlling the COVID-19 pandemic. Current global data indicate that more than 173 total doses were administered per 100 people, more than 66 people were vaccinated with a complete primary series per 100 people, and 31.9 people were vaccinated with at least one booster or additional dose per 100 people [10].

Serbia was the second country in the World Health Organization (WHO) European region to begin vaccinating against COVID-19, slightly later than the United Kingdom [11]. The Agency for Medicines and Medical Devices of Serbia registered four vaccines and approved their use: HB02 (Sinopharm), BNT162b2 (Pfizer-BioNTech), Gam-COVID-Vac (Sputnik-V), AZD1222 (AstraZeneca). HB02 was the most widely used in the Serbian vaccination campaign [12]. Serbian National Immunization Technical Advisory Groups (NITAG) developed recommendations on COVID-19 vaccination according to recommendations from leading organizations such as the WHO’s Strategic Advisory Group of Experts on Immunization and the European Technical Advisory Group of Experts on Immunizations, as well as the European Centre for Disease Prevention and Control (ECDC), the United Kingdom’s Joint Committee on Vaccination and Immunization, the United States (US) Advisory Committee on Immunization Practices, and articles published on Ministry of Health websites or in peer-reviewed journals [11]. Vaccination has been free of charge and voluntary. However, the coverage of vaccination in Serbia was modest, with 97.1 total doses per 100 people and 45.56 people were vaccinated with a complete primary series per 100 people but there are no official data about the number of people vaccinated with at least one booster or additional dose per 100 people [10].

Cooper et al. point out that hospitalised patients are especially vulnerable to COVID-19-associated complications and that controlling the transmission of COVID-19 in healthcare settings is critical to reduce the burden of this infection. They also emphasise the lack of studies that consider the effectiveness of vaccines (VE) against COVID-19 in hospitalized patients [13]. In order to generate real-life evidence on the safety and effectiveness of vaccines in use, the ECDC and the European Medical Agency officially established and launched consistent protocols to design necessary studies [14].

Our study aimed to estimate the VE of four different vaccine types and their combinations as primary or booster vaccines against SARS-CoV-2 infection in a population of hospitalized patients in a tertiary healthcare centre in Serbia during the Alpha, Delta, and Omicron waves.

## 2. Materials and Methods

### 2.1. Study Design and Settings

This retrospective cohort study was conducted at the Military Medical Academy (MMA), Belgrade, a 1000-bed tertiary healthcare centre. The MMA took care of adult (≥18 years) insured civilians from the whole territory of Serbia during the COVID-19 pandemic and cared for members of the military system as usual.

### 2.2. Study Population

The study kept track of a cohort of patients hospitalized in the MMA during the Alpha, Delta, and Omicron periods in Serbia. Through regular hospital surveillance for healthcare-associated infection, we identified hospitalized patients with symptoms of COVID-19 or who came into contact with known COVID-19 cases within 14 days during the study period between 15 February 2021 and 31 December 2022. During the observed period, the condition for admitting a patient to MMA, as well as in all hospitals in Serbia, was a negative antigen/PCR test 48 h up to admission to the hospital.

This work was completed as part of the COVID-19 outbreak operational evaluation in accordance with the recommendations of the Ministry of Health of the Republic of Serbia and the National Institute of Public Health of the Republic of Serbia [15]. During epidemiologic data collection, all patients were orally informed about PCR testing and the purpose of data collection and provided oral consent.

As in our previous survey, conducted during 2020, [16] epidemiological data on the following variables were gathered: demographic data (sex, age), exposure risk factors (treatment in a hospital with COVID-19 cases, contact with a known COVID-19 case within the previous 14 days), clinical signs and symptoms (fever, sore throat, cough, headache, myalgia/arthralgia, fatigue, gastrointestinal symptoms (nausea/vomiting/diarrhoea), and pneumonia (chest X-rays or computed tomography)), and data about comorbidities (no chronic diseases, chronic cardiac disease, cardiomyopathies, hypertension, chronic pulmonary diseases, chronic liver disease, diabetes mellitus, neurological diseases, malignancy, immunodeficiency, chronic kidney disease). The medical technician entered the collected data into a specially created access database daily, under the control of hospital epidemiologist.

Also, data about vaccinations were collected. All administered COVID-19 vaccines were registered in Serbian Vaccination Registry (SVR). Information about the vaccination products was retrieved from the SVR for every patient included in the study. Individuals were classified as fully vaccinated if at least 14 days had passed since the administration of the second dose of the HB02 (Sinopharm), BNT162b2 (Pfizer-BioNTech), Gam-COVID-Vac (Sputnik-V), or AZD1222 (AstraZeneca) according to national recommendations [17]. Also, under the administration of the first booster dose of the vaccine, it is assumed that the patient received the vaccine at least three months after the two doses in primary vaccination (the choice of the type of vaccine for the first-dose vaccine was free). Administration of the second booster dose meant that the patient received the vaccine at least five months after the administration of the first booster dose (choice of the type of vaccine for the second-dose vaccine was free). Time since vaccination was classified into three categories as follows: from time 0 up to ≤84 days after time 0; days 85–168, both included, after time 0; ≥169 days after time 0. Time 0 was defined as day 14 after the date of administration of the last vaccine dose [14]. Due to the small number of partially vaccinated patients, we excluded them from the study.

We defined a patient with a positive PCR result as an individual who tested with a positive PCR test result for SARS-CoV-2 performed on oro- and nasopharyngeal swabs and/or on respiratory-tract secretions and aspirates. Testing was carried out at Institute of Microbiology MMA. The presence of SARS-CoV-2 was detected by use of the “SARS-CoV-2 PLUS ELITe MGB^®^ Kit” (ELITechGroup SAS, Puteaux, France), a qualitative multiplex nucleic acids reverse transcription and amplification assay for the detection and identification of the RNA of Severe Acute Respiratory Syndrome Coronavirus 2 (SARS-CoV-2, ORF8 and ORF1ab genes), Influenza A Virus (FluA), Influenza B Virus (FluB), Respiratory Syncytial Virus type A and type B (RSV), in combination with ELITe InGenius^®^ (ELITechGroup SAS, Puteaux, France), a fully automated sample-to-result solution. We also used the Xpert Xpress SARS-CoV-2 test is an automated in vitro diagnostic test for qualitative detection of nucleic acid from SARS-CoV-2 (E (envelope) and N2 (nucleocapsid) genes). The Xpert Xpress SARS-CoV-2 test is performed on GeneXpert Instrument Systems (Gene Expert Dx system—Cepheid, Sunnyvale, CA, USA). As a detection sample we used a nasopharyngeal swab collected in UTM (Copan^®^ UTM^®^ Universal Transport Medium, 3 mL) (Copan^®^ UTM^®^ Universal Transport Medium, 3 mL—COPAN Diagnostic, Murrieta, CA, USA), transported and stored at room temperature (+18/+25 °C) for a maximum of 24 h or at +2/+8 °C for a maximum of five days, collected from individuals suspected of having COVID-19. The dates of the circulation of certain virus variants are determined based on Global Initiative on Sharing Avian Influenza Data—GISAID. Based on available information for the Republic of Serbia the dominance of the Alpha variant was in the period from 15 February 2021 until 21 June 2021; dominance of the Delta variant in the period from 21 June 2021 until 20 December 2021; and Omicron variants in the period from 20 December 2021 until 31 December 2022 [18].

The study was officially approved by The Ethics Committee of Military Medical Academy (N28/2021, 3 December 2021). This work was supported by the Ministry of Defence of the Republic of Serbia (grant number MF VMA 02/23-25).

### 2.3. Statistical Analysis

Descriptive and analytical statistical methods were used in data processing. Data are presented as mean ± standard deviation (SD) or median and number (percentage) for categorical variables. A chi-squared or Fisher’s exact test was used for the analysis of categorical data, as appropriate. VE was defined as 1-odds ratio (OR) obtained through univariate and multivariable logistic regression models for the outcome. The multivariable logistic regression model was used to derive adjusted OR with 95% confidence intervals (CI) for outcome which was adjusted for sex, age, number of comorbidities, and dominant strain. Outcome was defined as laboratory-confirmed SARS-CoV-2 infection during treatment in MMA (positive SARS-CoV-2 RT-PCR test). VE was calculated for hospitalized patients who were fully vaccinated with two doses of primary vaccination or with an additional booster dose received compared to non-vaccinated individuals [14]. Due to the small number of patients vaccinated with a second booster dose, they were assigned to the appropriate category with a single booster dose. The statistical analyses were performed using SPSS version 23.0 software (SPSS Inc., Chicago, IL, USA).

## 3. Results

The total number of patients tested within the cohort was 3593. During the Alpha strain’s dominance, out of 773 patients, 110 of them (14.2%) tested positive for SARS-CoV-2. In the duration of the Delta strain, there were slightly fewer positives at 11.7% (107 patients) of 909 patients tested during the Delta wave. During the Omicron strain we saw the largest number of patients tested, with a total of 1911 individuals, of which 252 (13.2%) tested positive.

Table 1 presents the demographic and clinical characteristics of the patients during the Alpha, Delta, and Omicron waves. Notably, there was no statistically significant difference in COVID-19 incidence by gender during all three waves. However, it was statistically significant that patients who had been treated in hospitals with confirmed COVID-19 cases or who had been in contact with COVID-19 patients were more likely to test positive during all three waves.

Concerning symptoms, fever was notably more prevalent in patients who tested positive during the Delta and Omicron waves. Sore throat was more common in positive patients during the Alpha and Omicron waves, while cough as a symptom was more common among positive patients in all three waves. Additionally, headache and myalgia/arthralgia were statistically significantly more frequent in positive patients during the Delta and Omicron waves.

It is worth mentioning that patients without chronic diseases were more prevalent among those who tested negative during the Omicron wave (8.7% among the negative group versus 3.6% among the positive group). Conversely, patients with chronic cardiac diseases were more common among those who tested positive during the Omicron wave. Regarding other comorbidities, there were no statistically significant differences. There were more patients with a previously documented COVID-19 infection during the Alpha wave among patients who tested positive and during the Omicron wave among patients who tested negative.

During the Alpha wave, patients who had received two doses of the COVID-19 vaccine were more likely to test negative. In terms of vaccination and contracting COVID-19, there were no statistically significant differences during the Delta and Omicron waves.

Out of the total cohort, 1866 (51.9%) patients had been vaccinated. Most of them received two doses of the Sinopharm vaccine, totalling 682 (36.5%), followed by three doses of the Sinopharm vaccine, accounting for 585 (31.4%). Additionally, 135 (7.2%) patients had received two doses of the Pfizer-BioNTech vaccine, and 73 (3.9%) had received three doses of the Pfizer-BioNTech vaccine. The most common combination of vaccines was two doses of the Sinopharm vaccine and a Pfizer-BioNTech booster, received by 148 (7.9%) respondents. The number of vaccines administered, based on doses and combinations, is displayed in Figure 1.

The VE of the Sinopharm vaccine after full vaccination, within the first 84 days after vaccination, was observed to be 45.6% (ranging from 68.6% to 5.8%). Having at least one Sinopharm booster, in the period more than 168 days after vaccination, demonstrated an effectiveness of 34.2%. At least one Pfizer-BioNTech booster following full vaccination with Sinopharm vaccination, showed a VE of 71.2% (ranging from 88.6% to 27.4%) in the period more than 168 days after vaccination. Table 2 illustrates the VE of various vaccine combinations based on the time elapsed since vaccination.

## 4. Discussion

The COVID-19 pandemic showed how big a challenge it was to organize the treatment of hospitalized patients without in-hospital transmission of SARS-CoV–2 [19], which is still a challenge now [20]. A national data linkage study from England showed that up to one in six SARS-CoV-2 infections among hospitalised patients with COVID-19 during the first half of the first year of the pandemic could be attributed to transmission in hospital [21]. A cohort study of US hospitals found that higher hospital-onset infection rates were associated with increases in community-onset SARS-CoV-2 infection rates according to the period of the COVID-19 pandemic, the admission testing rate, Census region, and the number of beds in hospital [22]. Vaccination as a measure of prevention for COVID-19 is still insufficiently researched in hospitalized patients, especially in countries with limited resources [23]. Our objective was to assess the VE of four different vaccine types and their combinations after full vaccination with two doses or after booster vaccines against SARS-CoV-2 infection in a population of hospitalized patients in a tertiary healthcare centre in Serbia during the Alpha, Delta, and Omicron waves. Also, we investigated the characteristics of the tested hospitalized patients, and their comorbidities during the mentioned periods.

During the COVID-19 pandemic, the symptom profiles varied across different virus variants and were influenced by vaccination. Initial research focused on hospitalized patients, but later, broader studies, such as a mobile app-based survey involving over two million people in the UK and the US, provided more generalizable data [24]. In the UK, notable symptoms among app participants included loss of smell and taste (64.76% in positive vs. 22.68% in negatives), fever (34.34% vs. 23.93%), skipped meals (25.95% vs. 19.24%), and diarrhoea (42.03% vs. 24.93%) [25]. Our study, focusing on already hospitalized patients, identified various non-specific symptoms like fever, headache, and muscle/joint pain, noting that these could also be present in other illnesses. Headache was reported in both COVID-19-positive and -negative individuals [26].

In our research, a sore throat was notably common during the Alpha and Omicron variants among our patients, although it is anticipated to occur across all COVID-19 variants. While the Omicron variant showed a reduction in respiratory symptoms [27], it was associated with cold-like symptoms that significantly affected daily activities and varied across its subvariants [28]. The variation in our study’s results may stem from examining patients at different disease stages and the subjective nature of symptom reporting, alongside differences between hospitalized patients with comorbidities and the general population.

A cough was identified as a consistent symptom across all variants in our study, echoing findings from Japan where coughing was highlighted as a predominant symptom [29]. A dry cough, reported by nearly 60% of SARS-CoV-2 positive individuals, was not limited to a specific variant [26]. In Canada, coughs was more prevalent among children with the Omicron and Delta variants compared to the Alpha variant, reflecting the evolution of the virus’s impact on children as the pandemic progressed [30].

Loss of the sense of taste and smell was dominant during the Alpha variant [31]. This symptom was dominant during the beginning of the pandemic, and as new variants appeared, the frequency decreased. In some studies, about two-thirds of positive patients reported impairment or loss of the sense of taste and smell during the Alpha and Delta variants, while during the Omicron variant, this percentage dropped to about 23.8% [26].

The clinical impact of SARS-CoV-2 infection varies widely, from asymptomatic cases to severe pneumonia leading to death, and is influenced by factors such as age, overall health, and comorbidities. Early in the pandemic, common comorbidities among infected patients included hypertension (21.1%), diabetes (9.7%), cardiovascular diseases (8.4%), and respiratory diseases (1.5%), which were associated with worse outcomes [32]. However, a study by Arshad et al., covering May 2020 to October 2021, found that comorbidities did not significantly affect COVID-19 severity in Pakistan [33]. Our findings indicated a higher proportion of patients without comorbidities among those testing negative for COVID-19 during the Omicron surge (8.7% vs. 3.6%, *p* = 0.008), but no significant difference was observed during the Alpha and Delta waves.

Momtazmanesh et al. reported a special association between cardiovascular diseases and COVID-19. More precisely, acute cardiac injury occurred in more than 25% of cases and mortality was 20 times higher in these patients [34]. During the Omicron wave, a significantly higher number of our patients with positive PCR tests had chronic cardiac diseases compared to negative patients (19% vs. 28%, *p* = 0.001). Also, the frequencies of our patients having hypertension and positive PCR tests during the Alpha, Delta, and Omicron waves were 35.5%, 49.5%, and 47.2%, respectively. Most authors show a lower frequency (21.1%; 22.9%), although hypertension with diabetes was identified as the most common comorbidity [32,35]. The higher frequency of hypertension in our cohort can be explained by the fact that almost half of the population over the age of 15 in Serbia suffers from hypertension [36].

Similar to hypertension, diabetes mellitus represents a health problem in Serbia. According to the Institute of Public Health of Serbia, 12.2% of the adult population suffers from diabetes [37] Thus, it is not surprising that the higher frequency of DM in the population of our infected patients during all three waves (16.4%, 21.5%, 17.9%) compared to that described in the literature [32,35].

Patients with comorbidities should be monitored during the course of COVID-19 due to the possible manifestation of a severe form of infection. Bearing in mind the severe clinical presentation and multisystem damage in these patients, it is necessary to carry out vaccination to prevent infection with SARS-CoV-2, as well as poor outcomes, as suggested by other authors [32,38].

Early VE for Sinopharm, Pfizer-BioNTech, Sputnik-V, and AstraZeneca against symptomatic, mild, and severe COVID-19 among individuals aged ≥60 in Vojvodina, Serbia, from January to April 2021 showed an overall VE of 88.4%, with variations among the vaccines: 86.9% for Sinopharm, 95% for Sputnik-V, and 99% for Pfizer-BioNTech. The study’s main limitation was its focus on the Alpha variant, excluding others like Delta or Omicron [39]. A separate study at the Royal Adelaide Hospital found that 24% of inpatients had not received any COVID-19 vaccine doses, indicating suboptimal vaccine uptake among a high-risk group [40]. In our cohort, more than 48% of patients had not been vaccinated, attributed to concerns over side effects, effectiveness, insufficient testing, mistrust in authorities, and conspiracy theories [41,42].

The demonstrated VE of the Pfizer-BioNTech vaccine emerged as the highest following full vaccination in our hospitalised patients. Its adjusted VE reached 74.8% (95% CI: −88.9–96.6) within 0–84 days after the second dose of the vaccine. However, protection waned over time dropping to 54.6% (95% CI: −94.1–89.4) at 85–168 days and subsequently reaching 19.8% (95% CI: −64.7–60.9) >168 days after the second dose. Gram et al. similarly found a decrease, with slightly higher values, in VE against symptomatic SARS-CoV-2 infection for both mRNA vaccines (Pfizer-BioNTech and Moderna vaccine) among individuals aged 60 and above in a Danish nationwide cohort study. They observed a VE against SARS Co 2 with the Alpha, Delta and Omicron, 14 to 30 days after the second dose of 90.7% (95% CI: 88.2–92.7), 82.3% (95% CI: 75.5–87.2), 39.9% (95%CI: 26.3–50.9), respectively. Also, they observed a VE against the Alpha, Delta, and Omicron SARS-CoV-2 variants, >120 days after the second dose, of 73.2% (95% CI: 57.1–83.3), 50.0% (95% CI: 46.7–53.0), 4.4% (95% CI: −0.1–8.7), respectively [43].

A test-negative study conducted in adults aged 50 years and over in France, between June 6, 2021 and February 10, 2022, reported a VE against symptomatic infection after 2-doses of vaccination (provided by the COVID-19 mRNA vaccines Pfizer-BioNTech, and Moderna) of 86% (95% CI: 75–92%) for Delta and 70% (95% CI: 58–79%) for Omicron, 7–30 days post vaccination. In addition, Franch’s study found that protection waned over time, reaching 60% (95% CI: 57–63%) against Delta and 20% (95% CI: 16–24%) for Omicron BA.1 > 120 days after vaccination [44].

Our assessment of the effectiveness of combinations involving a Pfizer-BioNTech or Sinopharm booster subsequent to full vaccination with Pfizer-BioNTech was hindered by the limited participant cohort receiving these specific regimens. However, a Hungarian study reported, in 65–100-years-old cohort, a lower crude incidence rate of SARS CoV-2 infection 14–120 days after full vaccination for each vaccine type (Pfizer-BioNTech, Moderna, Sputnik-V, AstraZeneca, Sinopharm, and Janssen) compared to the unvaccinated cohort (54.8 per 100,000 person-days), an increasing infection rate after 4 months, and very low rates (<10 per 100,000 people–days) after booster with mRNA regardless of primary vaccine type [8].

The demonstrated vaccine effectiveness of the Pfizer-BioNTech vaccine emerged as the highest following full vaccination, aligning with existing data where mRNA vaccines showed high vaccine effectiveness [45].

The Sinopharm vaccine’s effectiveness after full vaccination was observed to be 45.6% within the first 84 days after the second dose, slightly better than the 10% effectiveness reported in Hungary 14–120 days post-second dose [8]. In the UAE, a 45.6% breakthrough infection rate was noted for those with two Sinopharm doses plus a booster, the highest among examined regimens. In contrast, a two-dose Pfizer-BioNTech series plus booster showed a lower breakthrough rate of 29.4%. Mixed vaccination strategies, combining Sinopharm with a Pfizer-BioNTech booster, resulted in breakthrough rates of 36.4% and 33.3% for different combinations [46]. The mRNA vaccines, such as Pfizer and Moderna, were found to provide better protection than AstraZeneca, Janssen, and Sinopharm [47,48].

Our study suggests that a mixed-dose vaccination approach, combining an inactivated viral vaccine (Sinopharm HB02) as the first two doses followed by a booster dose with an mRNA vaccine (Pfizer-BioNTech), may offer enhanced protection against SARS-CoV-2. This finding aligns with similar research, such as a study comparing the immunogenicity of a group receiving two doses of BNT vaccine to one receiving two doses of Sinopharm vaccine followed by a BNT booster, indicating higher humoral immunity in the latter group [49]. Another U.S. study comparing various prime–booster combinations of the Pfizer-BioNTech, Moderna, and Janssen vaccines concluded that both homologous and heterologous booster regimens were safe and immunogenic [50]. Hungarian studies showed that administering a third dose of Pfizer-BioNTech following two doses of the Sinopharm vaccine significantly enhanced both humoral and T cell-mediated immune responses, comparable to three doses of the Pfizer-BioNTech vaccine [8,51]. In the group where participants were boosted with the Pfizer-BioNTech vaccine following a two-dose Sinopharm vaccine, the cumulative IFNγ-positive T cell response was surprisingly much higher than in groups where participants were immunized with two doses of the Pfizer-BioNTech and boosted with either the Pfizer-BioNTech or Sinopharm vaccine. The most remarkable results were obtained in the group with two doses of Sinopharm vaccine + Pfizer-BioNTech booster, highlighting that the most effective method of using the inactivated virus vaccines is heterologous boosting [51]. A study from China found that heterologous boosting with an mRNA vaccine (CS-2034, CanSino, Shanghai, China) induced higher immune responses and protection against symptomatic SARS-CoV-2 Omicron infections compared with homologous boosting with BBIBP-CorV—Sinopharm vaccine [52]. The seroconversion rates of SARS-CoV-2-specific neutralizing antibody responses were significantly higher in the mRNA heterologous booster regimen compared to the homologous booster regimen [52].

Because of the limited number of participants vaccinated with the AstraZeneca and Sputnik vaccines we were not able to assess the EV of these vaccines. However, the effectiveness of these vaccines has already been demonstrated. Data from Yordan suggest that the Sputnik vaccine was the most effective in preventing severe COVID-19 infection [53].

In Serbia, Sinopharm was the most widely used vaccine, followed by Pfizer-BioNTech mainly because the Sinopharm vaccine was the most available. Many citizens opted for a combination of these two vaccines, particularly receiving a Pfizer booster after the initial Sinopharm vaccination. Several factors influenced vaccine choice, including the availability of different vaccines and individual preferences. It has been shown by another study from Serbia that, for most people who decided to receive the Sinopharm vaccine, the main reason was the fact that it was manufactured using a well-known technology (inactivated virus) [54].

Regarding different periods studies suggest that VE was higher in the Alpha- than the Delta-dominant period. The VE after full vaccination against hospitalization varied from 54–85%. Receiving a booster dose after being fully vaccinated increased VE up to 90% [55]. During the Omicron wave, being fully vaccinated against COVID-19 significantly lowered the risk of hospitalization and severe consequences in cases involving the Omicron variant. Receiving a booster dose further enhanced this protective benefit [56,57].

This study demonstrates strengths in its diverse evaluation of multiple COVID-19 vaccines, offering insights into Pfizer-BioNTech and Sinopharm vaccines effectivity both individually and in combination. The longitudinal approach provides valuable insights into temporal vaccine effectiveness, crucial for booster dose considerations.

The limitation of our study is that we did not include data on the loss or impairment of the sense of taste and smell. Another limitation of our study is that the dominance of individual variants was not confirmed by sequencing for every positive patient due to limited financial resources. Also, the fact that we did not taken into account periods of co-circulation between one variant and another, between June and August 2021 (Alpha and Delta variant) and between November and December 2021 (Delta and Omicron variant), was limitation of our research.

## 5. Conclusions

Despite the availability of vaccines, a considerable proportion of the hospitalized patient population remained unvaccinated (48.1%). The demonstrated vaccine effectiveness of the Pfizer-BioNTech vaccine emerged as the highest following full vaccination. However, our study found that protection waned over time for all vaccine types. A Pfizer-BioNTech booster following the two doses of Sinopharm vaccine showed a VE of 71.2% in the period more than 168 days after vaccination, while the Sinopharm booster demonstrated a VE of 34.2% in the same period. Symptoms varied among different variants, with certain symptoms like coughs persisting across variants. Ultimately, these insights underscore the dynamic nature of vaccine effectiveness, highlighting the need for booster strategies, and the crucial importance of addressing concerns to enhance vaccination rates and public health strategies.

## Figures and Tables

**Figure 1 vaccines-12-00211-f001:**
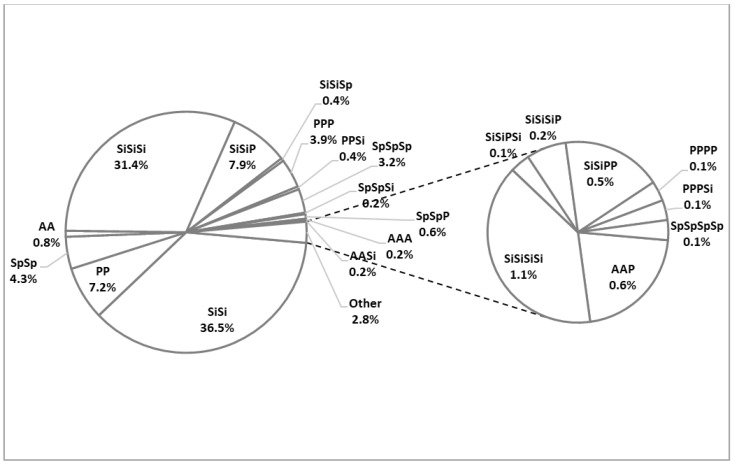
Patients by type end number of vaccine doses received. P: Pfizer-BionTech; Si: Sinopharm; Sp: Sputnik V; A: Astra Zeneca.

**Table 1 vaccines-12-00211-t001:** Characteristics of patients in cohort during the Alpha, Delta and Omicron waves.

	Negative Test (n = 663)	Positive Test (n = 110)	*p* Value	Negative Test (n = 802)	Positive Test (n = 107)	*p* Value	Negative Test (n = 1659)	Positive Test (n = 252)	*p* Value
	*Alfa*			*Delta*			*Omicron*		
Male. n (%) Female. n (%)	381 (87.0) 282 (84.2)	57 (13.0) 53 (15.8)	0.316	445 (55.5) 357 (44.5)	58 (54.2) 49 (45.8)	0.883	976 (58.8) 683 (41.2)	159 (63.1) 93 (36.9)	0.224
**Age, median (IQR)**	68 (21)	71 (21)	0.238	70 (21)	73.5 (24)	0.199	**68 (23)**	**72 (21)**	**0.007**
**Treating in hospital with COVID cases, n (%)**	**32 (12.7)**	**84 (29.1)**	**<0.001**	**46 (5.7)**	**30 (28.0)**	**<0.001**	**207 (12.5)**	**78 (31.0)**	**<0.001**
**Contact with known COVID case within 14 days, n (%)**	**80 (12.1)**	**31 (28.2)**	**<0.001**	**45 (5.6)**	**31 (29.0)**	**<0.001**	**181 (10.9)**	**78 (31.0)**	**<0.001**
Symptoms									
**Fever, n (%)**	243 (36.7)	49 (44.5)	0.140	**330 (41.1)**	**65 (60.7)**	**<0.001**	**579 (34.9)**	**125 (49.6)**	**<0.001**
**Sore throat, n (%)**	**18 (2.7)**	**8 (7.3)**	**0.030**	37 (4.6)	10 (9.3)	0.065	**81 (4.9)**	**38 (15.1)**	**<0.001**
**Cough, n (%)**	**118 (17.8)**	**29 (26.4)**	**0.047**	**133 (16.6)**	**35 (32.7)**	**<0.001**	**223 (13.4)**	**77 (30.6)**	**<0.001**
**Headache, n (%)**	47 (7.1)	10 (9.1)	0.584	**43 (5.4)**	**12 (11.2)**	**0.030**	**44 (2.7)**	**17 (6.7)**	**0.001**
**Myalgia/Arthralgia, n (%)**	54 (8.1)	15 (13.6)	0.091	**58 (7.2)**	**15 (14.0)**	**0.025**	**69 (4.2)**	**23 (9.1)**	**0.001**
**Fatigue, n (%)**	1 (0.2)	0 (0.0)	1.000	0 (0.0)	1 (0.9)	0.235	**4 (0.2)**	**4 (1.6)**	**0. 010**
Gastrointestinal symptoms, n (%)	0 (0.0)	0 (0.0)	/	0 (0.0)	1 (0.9)	0.235	1 (0.1)	0 (0.0)	1.000
Pneumonia, n (%)	172 (25.9)	22 (20.0)	0.225	187 (23.3)	33 (30.8)	0.113	252 (15.2)	35 (13.9)	0.657
Comorbidities									
**No chronic diseases, n (%)**	71 (10.7)	7 (6.4)	0.219	55 (6.9)	9 (8.4)	0.697	**144 (8.7)**	**9 (3.6)**	**0.008**
**Chronic cardiac disease, n (%)**	189 (28.55)	30 (27.3)	0.879	205 (25.6)	36 (33.6)	0.096	**316 (19.0)**	**71 (28.2)**	**0.001**
Hypertension, n (%)	273 (41.2)	39 (35.5)	0.304	325 (40.5)	53 (49.5)	0.095	715 (43.1)	119 (47.2)	0.245
Chronic pulmonary diseases, n (%)	38 (5.7)	8 (7.3)	0.678	49 (6.1)	5 (4.7)	0.709	99 (6.0)	15 (6.0)	1.000
Chronic liver diseases, n (%)	20 (3.0)	2 (1.8)	0.696	7 (0.9)	3 (2.8)	0.192	11 (0.7)	2 (0.8)	1.000
Diabetes mellitus, n (%)	98 (14.8)	18 (16.4)	0.775	131 (16.3)	23 (21.5)	0.230	269 (16.2)	45 (17.9)	0.572
Neurological diseases, n (%)	70 (10.6)	11 (10.0)	0.993	96 (12.0)	10 (9.3)	0.526	143 (8.6)	23 (9.1)	0.884
Malignancy, n (%)	152 (22.9)	25 (22.7)	1.000	186 (23.2)	27 (25.2)	0.729	448 (27.0)	59 (23.4)	0.260
Immunodeficiency, n (%)	2 (66.7)	1 (33.3)	0.369	1 (0.1)	0 (0.0)	1.000	5 (0.3)	0 (0.0)	1.000
Chronic kidney disease, n (%)	52 (7.8)	8 (7.3)	0.988	34 (4.2)	8 (7.5)	0.210	96 (5.8)	16 (6.3)	0.833
**Previous infection**	**113 (17.0)**	**61 (55.5)**	**<0.001**	105 (13.1)	14 (13.1)	0.998	**427 (25.7)**	**37 (14.7)**	**<0.001**
**Unvaccinated**	**512 (77.2)**	**99 (90.0)**	**0.003**	361 (45.0)	55 (51.4)	0.383	600 (36.2)	100 (39.7)	0.677
**Two doses**	**151 (22.8)**	**11 (10.0)**		330 (41.1)	41 (38.3)		333 (20.1)	46 (18.3)	
Three doses				111 (13.8)	11 (10.3)		690 (41.6)	102 (40.5)	
Four doses							36 (2.2)	4 (1.6)	

Bold: highlight the values.

**Table 2 vaccines-12-00211-t002:** Analysis of vaccine effectiveness: raw and adjusted data.

	n (%)	Crude Vaccine Effectivity % (95% CI)	Adjusted Vaccine Effectivity % (95% CI)
Unvaccinated	1727 (48.1)	Ref.	Ref.
**Pfizer-BionTech**			
Fully vaccinated 0–84	24 (0.67)	74.8 (−87.5–96.6)	74.8 (−88.9–96.6)
Fully vaccinated 85–168	31 (0.86)	60.0 (−68.6–90.5)	54.6 (−94.1–89.4)
Fully vaccinated >168	80 (2.23)	26.5 (−48.9–63.7)	19.8 (−64.7–60.9)
Fully vaccinated + at least one Pfizer booster 0–84	13 (0.36)	−5.4 (−378.5–76.8)	−19.8 (−450.6–73.9)
Fully vaccinated + at least one Pfizer booster 85–168	21 (0.58)	−36.5 (−308.8–54.5)	−30.5 (−297.1–57.1)
Fully vaccinated + at least one Pfizer booster >168	41 (1.58)	37.3 (−77.4–77.8)	40.8 (−69.6–79.4)
Fully vaccinated + at least one Sinopharm booster 0–84	1 (0.03)	NA	NA
Fully vaccinated + at least one Sinopharm booster 85–168	3 (0.08)	−190 (−3109.6–73.8)	−152.6 (−2725.9–77.4)
Fully vaccinated + at least one Sinopharm booster >168	4 (0.11)	−93.3 (−1765.7–80.0)	−74.9 (−1604.9–82.1)
**Sinopharm**			
Fully vaccinated 0–84	164 (4.56)	37.3 (−6.8–63.2)	**45.6 (5.8–68.6)**
Fully vaccinated 85–168	147 (4.09)	24.2 (−27.9–55.0)	28.9 (−23.8–59.1)
Fully vaccinated >168	371 (10.33)	19.9 (−12.3–43.0)	23.4 (−9.7–46.6)
Fully vaccinated + at least one Sinopharm booster 0–84	144 (4.01)	−4.6 (−67.8–34.8)	−1.8 (−66.5–37.8)
Fully vaccinated + at least one Sinopharm booster 85–168	106 (2.95)	−18.6 (−100.4–29.8)	−5.3 (−83.0–39.4)
Fully vaccinated + at least one Sinopharm booster >168	356 (9.91)	24.5 (−7.3–46.9)	**34.2 (3.0–55.4)**
Fully vaccinated + at least one Pfizer booster 0–84	29 (1.12)	57.0 (−81.8–89.8)	59.7 (−71.7–90.5)
Fully vaccinated + at least one Pfizer booster 85–168	38 (1.06)	−30.9 (−200.6–43,0)	−23.8 (−188.4–46.9)
Fully vaccinated + at least one Pfizer booster >168	95 (2.64)	**67.8 (19.9–87.0)**	**71.2 (27.4–88.6)**
**Sputnik**			
Fully vaccinated 0–84	17 (0.47)	63.8 (−174.5–95.2)	67.2 (−150.7–95.7)
Fully vaccinated 85–168	19 (0.53)	NA	NA
Fully vaccinated >168	45 (1.25)	27.5 (−85.4–71.7)	24.0 (−96.4–70.6)
Fully vaccinated + at least one Sputnik booster 0–84	15 (0.42)	10.8 (−297.7–80.0)	10.4 (−302.8–80.1)
Fully vaccinated + at least one Sputnik booster 85–168	16 (0.44)	−93.3 (−504.1–38.1)	−87.6 (−495.2–40.9)
Fully vaccinated + at least one Sputnik booster >168	31 (0.86)	60.0 (−68.6–90.5)	58.5 (−76.8–90.3)
**Astra-Zeneca**			
Fully vaccinated + at least one Fully vaccinated 0–84	4 (0.11)	−93.3 (−1765.7–80)	−144.0 (−2299.8–75.2)
Fully vaccinated + at least one Fully vaccinated 85–168	/	NA	NA
Fully vaccinated + at least one Fully vaccinated >168	10 (0.28)	35.6 (−410.8–91.9)	34.5 (−423.4–91.8)
Fully vaccinated + at least one Pfizer booster 0–84	8 (0.22)	17.2 (−576.2–89.9)	20.8 (−553.2–90.4)
Fully vaccinated + at least one Pfizer booster 85–168	1 (0.03)	NA	NA

Adjusted for sex, age, number of comorbidities, and dominant strain. Bold: highlight the values; NA: not available.

## Data Availability

The data presented in this study are available on request from the corresponding author. The data are not publicly available due to [privacy reasons].

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
