# Peer review of "Vaccine Effectiveness against SARS-CoV-2 Infection during the Circulation of Alpha, Delta, or Omicron Variants: A Retrospective Cohort Study in a Tertiary Hospital in Serbia"

_vaccines, 2024, doi:10.3390/vaccines12020211_

Round 1
Reviewer 1 Report
Comments and Suggestions for Authors
This manuscript written by Danijela Đurić–Petković etal hope to estimate the effectiveness of different vaccine types and their combinations as primary or booster vaccines against SARS-CoV-2 infection in a population of hospitalized patients in a healthcare centre in Serbia. They found some useful strategies of the use of vaccine. For example, mixed-dose strategies, Sinopharm as a primary dose followed by a Pfizer-BioNTech booster, suggested increased protection. Overall, I think this work provide knowledge about the vaccine effectiveness in Serbia and can be published to the public. My major concern is the data presentation.
1. too much data in the table. Key results and facts should be emphasized or highlighted.
2. Figure is necessary to better reflect the results, such as a pie chart for the data.
Comments on the Quality of English Language
Not bad
Author Response
This manuscript written by Danijela Đurić–Petković et al hopes to estimate the effectiveness of different vaccine types and their combinations as primary or booster vaccines against SARS-CoV-2 infection in a population of hospitalized patients in a healthcare centre in Serbia. They found some useful strategies for the use of vaccines. For example, mixed-dose strategies, Sinopharm as a primary dose followed by a Pfizer-BioNTech booster, suggested increased protection. Overall, I think this work provides knowledge about the vaccine effectiveness in Serbia and can be published to the public. My major concern is the data presentation.
We thank the Reviewer for their positive comment. Also, thank you for allowing us to submit a revised draft of our manuscript. We appreciate the time and effort that you dedicated to providing feedback on our manuscript and are grateful for the insightful comments and valuable improvements to our paper. We have been able to incorporate changes to reflect most of the suggestions. You can track changes according to your requirements.
- Too much data in the table. Key results and facts should be emphasized or highlighted.
Thank you for bringing this to our attention. In response, we have made sure to emphasize the key results within our tables by making them bold. This should make it easier to identify and understand the most significant findings. We hope this enhances the clarity and usability of our results.
- Figure is necessary to better reflect the results, such as a pie chart for the data.
Thank you for your comment. Based on your suggestion, we've carefully revised Table 2 in the Figure 1. We're grateful for your guidance and hope that this modification improves the clarity of our results.
Reviewer 2 Report
Comments and Suggestions for Authors
My largest concern with this manuscript is the conclusion that Sinopharm primary vaccination followed by Pfizer vaccine boost provides a positive boost to protection. My main reasons for this concern stem from the data presented in Table 3 of the manuscript.
1. The data for the Pfizer vaccine primary vaccination followed by Sinopharm booster has only 1-4 subject data. It is very hard to make any conclusions on the performance of this data with that very limited number of cases.
2. The data for Pfizer vaccine alone as primary shows 74.8% (96.6; -88.9) efficacy. The data for the combined Sinopharm primary vaccination followed by Pfizer vaccine boost shows 71.2% (88.6; 27.4) efficacy. There is simply no way that the authors can conclude that the vaccine shows greater efficacy by combining them as opposed to vaccinating with Pfizer vaccine alone. It is possible since this data is derived from the first exposure of these subjects to the Pfizer vaccine, that the effect at the boost is comparable to that seen with the primary vaccination subjects in that a high degree of efficacy is observed at this point of exposure.
3. A negative effect is actually observed when looking at subjects vaccinated with Sinopharm followed by a Pfizer boost at day 85-168, Pfizer booster 85-168 -23.8% (46.9; -188.4) efficacy. Can the authors explain why this effect would be seen at this interval but a benefit at the longer interval of > 168 days. This pattern seems random.
4. The risk factors associated with the longer period of time that subjects vaccinated with the Sinopharm vaccine experience until they get vaccinated with the Pfizer vaccine is considerable (45.6% (68.6; 5.8) for Sinopharm versus 74.8% (96.6; -88.9) for Pfizer as primary vaccination. Can the authors address this point as to why it would be preferable to utilize the Sinopharm vaccination as prime followed by Pfizer boost rather than utilizing the Pfizer primary vaccination to start with?
5. Table 3 also has Ref. listed for unvaccinated. Does this mean to refer to a reference that needs to be cited or what is the meaning of this entry in the table?
Author Response
My largest concern with this manuscript is the conclusion that Sinopharm primary vaccination followed by Pfizer vaccine boost provides a positive boost to protection. My main reasons for this concern stem from the data presented in Table 3 of the manuscript.
We thank the Reviewer for raising this critical issue. We agree that the conclusion that Sinopharm primary vaccination followed by Pfizer vaccine boost provides a positive boost to protection is not good. We agree that it is necessary to draw better conclusions. Thank you for allowing us to submit a revised draft of our manuscript. We appreciate the time and effort that you dedicated to providing feedback on our manuscript and are grateful for the insightful comments and valuable improvements to our paper. We have been able to incorporate changes to reflect most of the suggestions. You can track changes according to your requirements.
- The data for the Pfizer vaccine primary vaccination followed by Sinopharm booster has only 1-4 subject data. It is very hard to make any conclusions on the performance of this data with that very limited number of cases.
Thank you for pointing this out. In part of the Statistical analysis we have specified “ Due to the small number of patients vaccinated with a second booster dose, they were assigned to the appropriate category with a single booster dose.”
- The data for Pfizer vaccine alone as primary shows 74.8% (96.6; -88.9) efficacy. The data for the combined Sinopharm primary vaccination followed by Pfizer vaccine boost shows 71.2% (88.6; 27.4) efficacy. There is simply no way that the authors can conclude that the vaccine shows greater efficacy by combining them as opposed to vaccinating with Pfizer vaccine alone. It is possible since this data is derived from the first exposure of these subjects to the Pfizer vaccine, that the effect at the boost is comparable to that seen with the primary vaccination subjects in that a high degree of efficacy is observed at this point of exposure.
We thank the Reviewer for raising this critical issue.
We have added a new sentence in our Conclusion “Demonstrated vaccine effectiveness of the Pfizer-BioNTech vaccine emerged as the highest following primary vaccination.”
Also, we have excluded the sentence “Mixed-dose vaccination strategies, particularly using Sinopharm as the primary dose followed by a Pfizer-BioNTech booster, hinted at enhanced protection against SARS-CoV-2.”
Moreover, we have added one new sentences in our Conclusion:
“However, our study found that protection waned over time for all vaccine types. Pfizer-BioNTech booster following the primary Sinopharm vaccination, showed VE of 71.2% in the period more than 168 days after vaccination, while the Sinopharm booster demonstrated VE of 34.2% in the same period.”
We also deleted a sentence from the Abstract
”Combining Sinopharm and Pfizer-BioNTech vaccines hinted at enhanced protection against SARS-CoV-2”
- A negative effect is actually observed when looking at subjects vaccinated with Sinopharm followed by a Pfizer boost at day 85-168, Pfizer booster 85-168 -23.8% (46.9; -188.4) efficacy. Can the authors explain why this effect would be seen at this interval but a benefit at the longer interval of > 168 days. This pattern seems random.
We are grateful to the Reviewer for highlighting this important concern. While the pattern observed may appear random, we believe that it is primarily attributable to the larger number of patients in the group exceeding 168 days, which may have impacted the ability to achieve statistical significance. We appreciate the opportunity to clarify this aspect of our analysis.
- The risk factors associated with the longer period of time that subjects vaccinated with the Sinopharm vaccine experience until they get vaccinated with the Pfizer vaccine is considerable (45.6% (68.6; 5.8) for Sinopharm versus 74.8% (96.6; -88.9) for Pfizer as primary vaccination. Can the authors address this point as to why it would be preferable to utilize the Sinopharm vaccination as prime followed by Pfizer boost rather than utilizing the Pfizer primary vaccination to start with?
As the authors of this study, we tried to explain why Sinofarm was the most represented vaccine. We personally do not think that this vaccine is a better choice than RNA vaccines, but it has been procured and our citizens have decided to receive it. We are just analyzing what happened in our hospital after that
“In Serbia, Sinopharm was the most widely used vaccine, followed by Pfizer-BioNTech mainly because the Sinopharm vaccine was the most available vaccine. Many citizens opted for a combination of these two vaccines, particularly receiving a Pfizer booster after the initial Sinopharm vaccination. Several factors influenced vaccine choice, including the availability of different vaccines and individual preferences. It was shown by another study from Serbia that for most people who decided to receive the Sinopharm vaccine, the main reason was the fact that it was manufactured using a well-known technology (inactivated virus) [54].”
- Table 3 also has Ref. listed for unvaccinated. Does this mean to refer to a reference that needs to be cited or what is the meaning of this entry in the table?
Thank you for your comment. To clarify, "Ref." indicates that we used the unvaccinated group as the reference category for our comparisons. We hope this explanation helps to better understand the context of our analysis.
Reviewer 3 Report
Comments and Suggestions for Authors
Dear authors,
I read with particular interest manuscript entitled "Vaccine effectiveness against SARS-CoV-2 infection during circulation of Alpha, Delta, or Omicron variant: a retrospective cohort study in a tertiary hospital in Serbia", in which the authors estimate the effectiveness of four different vaccines against SARS-CoV-2 during the Alpha, Delta, and Omicron waves.
This paper is well written, but the authors must clarify some aspects of the manuscript.
Line 42: define the abbreviation COVID-19, as this is the first time it appears in the text. It has been described below in line 43. Other abbreviations appearing in the text could also be defined (SD, VE).
If I understand correctly, the criteria for inclusion in the study are hospitalized patients with a negative test for SARS-CoV-2 48 hours before admission and patients with symptoms of COVID-19 or who have been in contact with known cases of COVID-19. Is this the case? Did all the patients included have symptoms or have been in contact with a case? In terms of age, were patients of any age selected? Please clarify.
Only patients with a positive PCR result are defined in the study population section, but in the statistical analysis section, "positive SARS-CoV-2 RT-PCR or rapid antigen test" is mentioned. Was the rapid antigen test also used to diagnose patients? Please specify. Standardize PCR terms
The periods of dominance of the different variants were determined based on data from GISAID and the Republic of Serbia. However, it was not considered that there were periods of co-circulation between one variant and another, between June and August between Alpha and Delta and between November and December between Delta and Omicron. Was this fact taken into account?
Identification of the samples should be carried out; the test par excellence is sequencing, but there is also the option of PCR kits capable of discriminating between variants to get an idea of what is circulating in the different established periods.
The efficacy analysis included periods in which the vaccination status of the population was very different: during the Alpha period, low coverage, and in the Omicron period, higher coverage and higher number of vaccine doses, which is very important to take into account since efficacy estimates will be different. The rapid increase in vaccine coverage may affect comparability between periods. Analyses should be stratified by period and adjusted for vaccination status. When indicating the vaccination status, the authors use "primary" and "booster"; it is more correct to put "partially vaccinated", "fully vaccinated", and "fully vaccinated and the number of booster doses". Also, "first dose", "second dose", "first booster",...
A stratified analysis by age range could be considered since the vaccine's effectiveness may vary depending on the age group. It would also be interesting to estimate the effectiveness of the vaccines in terms of the periods compared between the unvaccinated and the vaccinated. An effectiveness analysis could be made according to age groups, chronic condition, presence or absence of symptoms and type of vaccine.
For example, I cite some articles related to the subject that have not been included in the bibliography:
Martínez-Baz I, Trobajo-Sanmartín C, Miqueleiz A, Casado I, Navascués A, Burgui C, Ezpeleta C, Castilla J, Guevara M; Working Group for the Study of COVID-19 in Navarra; Members of the Working Group for the Study of COVID-19 in Navarra. Risk reduction of hospitalization and severe disease in vaccinated COVID-19 cases during the SARS-CoV-2 variant Omicron BA.1-predominant period, Navarre, Spain, January to March 2022. Euro Surveill. 2023, 28(5), 2200337. doi: 10.2807/1560-7917.ES.2023.28.5.2200337.
Rose AM, Nicolay N, Sandonis Martín V, Mazagatos C, Petrović G, Niessen FA, Machado A, Launay O, Denayer S, Seyler L, Baruch J, Burgui C, Loghin II, Domegan L, Vaikutytė R, Husa P, Panagiotakopoulos G, Aouali N, Dürrwald R, Howard J, Pozo F, Sastre-Palou B, Nonković D, Knol MJ, Kislaya I, Luong Nguyen LB, Bossuyt N, Demuyser T, Džiugytė A, Martínez-Baz I, Popescu C, Duffy R, Kuliešė M, Součková L, Michelaki S, Simon M, Reiche J, Otero-Barrós MT, Lovrić Makarić Z, Bruijning-Verhagen PC, Gomez V, Lesieur Z, Barbezange C, Van Nedervelde E, Borg ML, Castilla J, Lazar M, O'Donnell J, Jonikaitė I, Demlová R, Amerali M, Wirtz G, Tolksdorf K, Valenciano M, Bacci S, Kissling E; I-MOVE-COVID-19 hospital study team; VEBIS hospital study team; Members of the I-MOVE-COVID-19 and VEBIS hospital study teams (in addition to the named authors). Vaccine effectiveness against COVID-19 hospitalization in adults (≥ 20 years) during Alpha- and Delta-dominant circulation: I-MOVE-COVID-19 and VEBIS SARI VE networks, Europe, 2021. Euro Surveill. 2023, 28(47), 2300186. doi: 10.2807/1560-7917.ES.2023.28.47.2300186.
Rose AM, Nicolay N, Sandonis Martín V, Mazagatos C, Petrović G, Baruch J, Denayer S, Seyler L, Domegan L, Launay O, Machado A, Burgui C, Vaikutyte R, Niessen FA, Loghin II, Husa P, Aouali N, Panagiotakopoulos G, Tolksdorf K, Horváth JK, Howard J, Pozo F, Gallardo V, Nonković D, Džiugytė A, Bossuyt N, Demuyser T, Duffy R, Luong Nguyen LB, Kislaya I, Martínez-Baz I, Gefenaite G, Knol MJ, Popescu C, Součková L, Simon M, Michelaki S, Reiche J, Ferenczi A, Delgado-Sanz C, Lovrić Makarić Z, Cauchi JP, Barbezange C, Van Nedervelde E, O'Donnell J, Durier C, Guiomar R, Castilla J, Jonikaite I, Bruijning-Verhagen PC, Lazar M, Demlová R, Wirtz G, Amerali M, Dürrwald R, Kunstár MP, Kissling E, Bacci S, Valenciano M; I-MOVE-COVID-19 hospital study team; VEBIS hospital study team; Members of the I-MOVE-COVID-19 and VEBIS hospital study teams (in addition to authors above). Vaccine effectiveness against COVID-19 hospitalization in adults (≥ 20 years) during Omicron-dominant circulation: I-MOVE-COVID-19 and VEBIS SARI VE networks, Europe, 2021 to 2022. Euro Surveill. 2023, 28(47), 2300187. doi: 10.2807/1560-7917.ES.2023.28.47.2300187.
In the results, put the CI values as in the discussion, 45.6% (95% CI: 5.8-68.6) and when putting the intervals, it is indicated from the lowest to the highest, and the semicolons should be removed; please modify both in the text and Table 3.
The discussion is extensive; much data from other articles is provided that needs to be discussed with the results obtained in this study. For example, in section 4.3 the effectiveness values obtained in this study are not compared with those of other articles. A more general discussion could be considered without the need for sections on the results obtained in this study.
Author Response
I read with particular interest manuscript entitled "Vaccine effectiveness against SARS-CoV-2 infection during circulation of Alpha, Delta, or Omicron variant: a retrospective cohort study in a tertiary hospital in Serbia", in which the authors estimate the effectiveness of four different vaccines against SARS-CoV-2 during the Alpha, Delta, and Omicron waves.
This paper is well written, but the authors must clarify some aspects of the manuscript.
We thank the Reviewer for their positive comment. Also, thank you for giving us the opportunity to submit a revised draft of our manuscript. We appreciate the time and effort that you dedicated to providing feedback on our manuscript and are grateful for the insightful comments and valuable improvements to our paper. We have been able to incorporate changes to reflect most of the suggestions. You can track changes according to your requirements.
Line 42: define the abbreviation COVID-19, as this is the first time it appears in the text. It has been described below in line 43. Other abbreviations appearing in the text could also be defined (SD, VE).
Thank you for this comment. We have defined the mentioned abbreviations (COVID-19, SD, VE) the first time they appear in the text
If I understand correctly, the criteria for inclusion in the study are hospitalized patients with a negative test for SARS-CoV-2 48 hours before admission and patients with symptoms of COVID-19 or who have been in contact with known cases of COVID-19. Is this the case? Did all the patients included have symptoms or have been in contact with a case? In terms of age, were patients of any age selected? Please clarify.
We thank the Reviewer 3 for this valuable comments. Describing our study population we pointed out that “Through regular hospital surveillance for healthcare-associated infection, we identified hospitalized patients with symptoms of COVID-19 or contact with known COVID-19 cases within 14 days during the study period between 15 February 2021 and 31 December 2022.”Also, we have noticed that all patients which treated in MMA in a period of study tested negative for SARS-CoV-2 48 hours before admission “During the observed period, the condition for admitting a patient to MMA, as well as in all hospitals in Serbia, was a negative antigen/PCR test 48 hours up to admission to the hospital.”
Also, we have specified the age of our hospitalized patients “The MMA is taking care of adult (≥ 18 years) insured civilians from the whole territory of Serbia during the COVID-19 pandemic and caring for members of the military system as usual.
Only patients with a positive PCR result are defined in the study population section, but in the statistical analysis section, "positive SARS-CoV-2 RT-PCR or rapid antigen test" is mentioned. Was the rapid antigen test also used to diagnose patients? Please specify. Standardize PCR terms
Thank you for pointing this out. We apologize for making a mistake. Only SARS-CoV-2 RT-PCR used to diagnose patients. We have excluded this part of the sentence “or rapid antigen test”
The periods of dominance of the different variants were determined based on data from GISAID and the Republic of Serbia. However, it was not considered that there were periods of co-circulation between one variant and another, between June and August between Alpha and Delta and between November and December between Delta and Omicron. Was this fact taken into account?
Thank you for this comment. We have defined that as limitation of our study. We have added sentences
“Another limitation of our study is that the dominance of individual variants was not confirmed by sequencing for every positive patient due to limited financial resources. Also fact that we were not taken in account periods of co-circulation between one variant and another, between June and August 2021 (Alpha and Delta variant) and between November and December 2021 (Delta and Omicron variant) was limitation of our research.”
Identification of the samples should be carried out; the test par excellence is sequencing, but there is also the option of PCR kits capable of discriminating between variants to get an idea of what is circulating in the different established periods.
Thank you for your valuable suggestion. We responded to this suggestion in the previous quote “Another limitation of our study is that the dominance of individual variants was not confirmed by sequencing for every positive patient due to limited financial resources.”
The efficacy analysis included periods in which the vaccination status of the population was very different: during the Alpha period, low coverage, and in the Omicron period, higher coverage and higher number of vaccine doses, which is very important to take into account since efficacy estimates will be different. The rapid increase in vaccine coverage may affect comparability between periods. Analyses should be stratified by period and adjusted for vaccination status. When indicating the vaccination status, the authors use "primary" and "booster"; it is more correct to put "partially vaccinated", "fully vaccinated", and "fully vaccinated and the number of booster doses". Also, "first dose", "second dose", "first booster",...
Thank you for bringing this to our attention. Following your suggestion, we have updated the categorization of vaccination status to distinguish between those who are fully vaccinated and those who have received the booster doses. We changed the vaccination status in fully vaccinated or fully vaccinated and the number of booster doses as you suggested. Given the limited number of patients who have received a second booster dose, we have thoughtfully grouped them with individuals who have received a single booster dose for a more cohesive analysis.
A stratified analysis by age range could be considered since the vaccine's effectiveness may vary depending on the age group. It would also be interesting to estimate the effectiveness of the vaccines in terms of the periods compared between the unvaccinated and the vaccinated. An effectiveness analysis could be made according to age groups, chronic condition, presence or absence of symptoms and type of vaccine.
Thank you for your valuable suggestion. We did explore the possibility of conducting a stratified analysis by periods. However, dividing the patient data into numerous groups made it challenging to derive statistically significant outcomes. This limitation was also encountered with other attempts at stratification. Given these constraints, we opted to refine our analysis by adjusting for variables such as sex, age, number of comorbidities, and the dominant strain. This approach aims to ensure a more focused and meaningful analysis. We appreciate your understanding and support as we navigate these complex analytical decisions.
For example, I cite some articles related to the subject that have not been included in the bibliography:
Martínez-Baz I, Trobajo-Sanmartín C, Miqueleiz A, Casado I, Navascués A, Burgui C, Ezpeleta C, Castilla J, Guevara M; Working Group for the Study of COVID-19 in Navarra; Members of the Working Group for the Study of COVID-19 in Navarra. Risk reduction of hospitalization and severe disease in vaccinated COVID-19 cases during the SARS-CoV-2 variant Omicron BA.1-predominant period, Navarre, Spain, January to March 2022. Euro Surveill. 2023, 28(5), 2200337. doi: 10.2807/1560-7917.ES.2023.28.5.2200337.
Rose AM, Nicolay N, Sandonis Martín V, Mazagatos C, Petrović G, Niessen FA, Machado A, Launay O, Denayer S, Seyler L, Baruch J, Burgui C, Loghin II, Domegan L, Vaikutytė R, Husa P, Panagiotakopoulos G, Aouali N, Dürrwald R, Howard J, Pozo F, Sastre-Palou B, Nonković D, Knol MJ, Kislaya I, Luong Nguyen LB, Bossuyt N, Demuyser T, Džiugytė A, Martínez-Baz I, Popescu C, Duffy R, Kuliešė M, Součková L, Michelaki S, Simon M, Reiche J, Otero-Barrós MT, Lovrić Makarić Z, Bruijning-Verhagen PC, Gomez V, Lesieur Z, Barbezange C, Van Nedervelde E, Borg ML, Castilla J, Lazar M, O'Donnell J, Jonikaitė I, Demlová R, Amerali M, Wirtz G, Tolksdorf K, Valenciano M, Bacci S, Kissling E; I-MOVE-COVID-19 hospital study team; VEBIS hospital study team; Members of the I-MOVE-COVID-19 and VEBIS hospital study teams (in addition to the named authors). Vaccine effectiveness against COVID-19 hospitalization in adults (≥ 20 years) during Alpha- and Delta-dominant circulation: I-MOVE-COVID-19 and VEBIS SARI VE networks, Europe, 2021. Euro Surveill. 2023, 28(47), 2300186. doi: 10.2807/1560-7917.ES.2023.28.47.2300186.
Rose AM, Nicolay N, Sandonis Martín V, Mazagatos C, Petrović G, Baruch J, Denayer S, Seyler L, Domegan L, Launay O, Machado A, Burgui C, Vaikutyte R, Niessen FA, Loghin II, Husa P, Aouali N, Panagiotakopoulos G, Tolksdorf K, Horváth JK, Howard J, Pozo F, Gallardo V, Nonković D, Džiugytė A, Bossuyt N, Demuyser T, Duffy R, Luong Nguyen LB, Kislaya I, Martínez-Baz I, Gefenaite G, Knol MJ, Popescu C, Součková L, Simon M, Michelaki S, Reiche J, Ferenczi A, Delgado-Sanz C, Lovrić Makarić Z, Cauchi JP, Barbezange C, Van Nedervelde E, O'Donnell J, Durier C, Guiomar R, Castilla J, Jonikaite I, Bruijning-Verhagen PC, Lazar M, Demlová R, Wirtz G, Amerali M, Dürrwald R, Kunstár MP, Kissling E, Bacci S, Valenciano M; I-MOVE-COVID-19 hospital study team; VEBIS hospital study team; Members of the I-MOVE-COVID-19 and VEBIS hospital study teams (in addition to authors above). Vaccine effectiveness against COVID-19 hospitalization in adults (≥ 20 years) during Omicron-dominant circulation: I-MOVE-COVID-19 and VEBIS SARI VE networks, Europe, 2021 to 2022. Euro Surveill. 2023, 28(47), 2300187. doi: 10.2807/1560-7917.ES.2023.28.47.2300187.
Thank you for your valuable suggestion. In response, we have thoughtfully expanded our list of references to include the ones you suggested. We appreciate your contribution to enriching the depth of our research.
In the results, put the CI values as in the discussion, 45.6% (95% CI: 5.8-68.6) and when putting the intervals, it is indicated from the lowest to the highest, and the semicolons should be removed; please modify both in the text and Table 3.
Thank you for this valuable suggestion. We have carefully adjusted the Confidence Interval (CI) values in line with your recommendation.
The discussion is extensive; much data from other articles is provided that needs to be discussed with the results obtained in this study. For example, in section 4.3 the effectiveness values obtained in this study are not compared with those of other articles. A more general discussion could be considered without the need for sections on the results obtained in this study.
Thank you for highlighting this aspect. In response, we've shortened the discussion section, ensuring it is now presented as a unified whole without separate parts. We appreciate your constructive feedback.
Round 2
Reviewer 2 Report
Comments and Suggestions for Authors
The questions and concerns raised in my initial review have been addressed by the authors. I believe that the manuscript is suitable for publication.
Author Response
We sincerely appreciate the effort you've put into evaluating our manuscript and offering constructive suggestions for improvement. Thank you once again for your valuable contribution to our research process.
Reviewer 3 Report
Comments and Suggestions for Authors
Dear authors,
First, thank you for replying to the comments in the first review. Many of the doubts have been solved and corrected. However, I would like to make some suggestions;
As I said in the previous review, I consider the discussion too long; it could be more summarized.
The figure option is more accurate than the table, but have you considered making a pie chart instead of a bar chart?
Is it not possible to stratify by age range?
Are there no patients who have only received a single dose of vaccine (partially vaccinated)?
Thank you
Author Response
Dear authors,
First, thank you for replying to the comments in the first review. Many of the doubts have been solved and corrected. However, I would like to make some suggestions;
We thank the Reviewer for taking the time to review our work and provide feedback. We sincerely appreciate the effort you've put into evaluating our manuscript and offering constructive suggestions for improvement. We've carefully considered each of your recommendations and have made efforts to address them in the revised manuscript. We believe these changes have significantly strengthened our work and hope you find the revisions satisfactory. Thank you once again for your valuable contribution to our research process.
As I said in the previous review, I consider the discussion too long; it could be more summarized.
Thank you for your advice. Following your suggestion, we have taken steps to condense the discussion section of our manuscript. This time our corrections are highlighted in light blue in the text of the manuscript.
The figure option is more accurate than the table, but have you considered making a pie chart instead of a bar chart?
Thank you for your valuable suggestion. Following your advice, we have transitioned from using bar chart to pie chart in our presentation of the data.
Is it not possible to stratify by age range?
Thank you for your suggestion regarding conducting a stratified analysis by age range. We encountered a challenge in dividing the patient data into multiple age groups, vaccine type and time elapsed since vaccination, which resulted in a lot of groups having small numbers. This fragmentation made it difficult to obtain statistically meaningful results across the various categories. We acknowledge the importance of stratified analysis in understanding nuanced differences among patient groups, but in this case, it was not possible. We hope you understand the rationale behind our decision.
Are there no patients who have only received a single dose of vaccine (partially vaccinated)?
The number of these patients was notably small, which led us to the decision not to include them in our analysis. This decision was made with careful consideration, as we aimed to ensure the robustness and statistical significance of our findings. We understand the value of including a diverse patient population in research; however, in this instance, we decided to exclude these patients. We appreciate your understanding of the complexities involved in our study design and the thoughtful decisions we've had to make. This sentence was added into the methodology section to clarity: Due to the small number of partially vaccinated patients, we excluded them from the study.